# Treatment of Ready-To-Eat Cooked Meat Products with Cold Atmospheric Plasma to Inactivate *Listeria* and *Escherichia coli*

**DOI:** 10.3390/foods12040685

**Published:** 2023-02-04

**Authors:** Isabella Csadek, Ute Vankat, Julia Schrei, Michelle Graf, Susanne Bauer, Brigitte Pilz, Karin Schwaiger, Frans J. M. Smulders, Peter Paulsen

**Affiliations:** Unit of Food Hygiene and Technology, Institute of Food Safety, Food Technology and Veterinary Public Health, University of Veterinary Medicine, Veterinaerplatz 1, 1210 Vienna, Austria

**Keywords:** cold atmospheric plasma, dielectric barrier discharge, antimicrobial effects, *Listeria*, *Escherichia coli*, cooked cured meat products, colour

## Abstract

Ready-to-eat meat products have been identified as a potential vehicle for *Listeria monocytogenes*. Postprocessing contamination (i.e., handling during portioning and packaging) can occur, and subsequent cold storage together with a demand for products with long shelf life can create a hazardous scenario. Good hygienic practice is augmented by intervention measures in controlling post-processing contamination. Among these interventions, the application of ‘cold atmospheric plasma’ (CAP) has gained interest. The reactive plasma species exert some antibacterial effect, but can also alter the food matrix. We studied the effect of CAP generated from air in a surface barrier discharge system (power densities 0.48 and 0.67 W/cm^2^) with an electrode-sample distance of 15 mm on sliced, cured, cooked ham and sausage (two brands each), veal pie, and calf liver pâté. Colour of samples was tested immediately before and after CAP exposure. CAP exposure for 5 min effectuated only minor colour changes (ΔE max. 2.7), due to a decrease in redness (a*), and in some cases, an increase in b*. A second set of samples was contaminated with *Listeria (L.) monocytogenes*, *L. innocua* and *E. coli* and then exposed to CAP for 5 min. In cooked cured meats, CAP was more effective in inactivating *E. coli* (1 to 3 log cycles) than *Listeria* (from 0.2 to max. 1.5 log cycles). In (non-cured) veal pie and calf liver pâté that had been stored 24 h after CAP exposure, numbers of *E. coli* were not significantly reduced. Levels of *Listeria* were significantly reduced in veal pie that had been stored for 24 h (at a level of ca. 0.5 log cycles), but not in calf liver pâté. Antibacterial activity differed between but also within sample types, which requires further studies.

## 1. Introduction

*Listeria* (*L*.) *monocytogenes* is an important food-borne pathogen and can thrive and persist in a wide range of environmental conditions, even under industrial conditions in food processing companies [1]. Asymptomatic ‘healthy’ animals and humans may carry and shed the pathogen [2,3]. However, clinical symptoms may develop and range from mild fever to severe diarrhoeal disease, fatalities or even miscarriages, with young, old, and immunocompromised consumers at particular risk. Predominant symptoms are not necessarily specific, e.g., chills, headache, arthralgia, prostration, malaise, swollen lymph nodes [4].

Food intended for human consumption can be contaminated with *L. monocytogenes* at virtually any level in the food chain, i.e., primary production at the farm level, during processing or at the retail or consumer level due to insufficient hygiene precautions [3]. As early as 1983, Schlech et al. reported transmission of the bacterium via food [5]. *L. monocytogenes* is considered the most important food-borne pathogen in ready-to-eat (RTE) foods due to its ability to survive and multiply under cold storage conditions, in vacuum or modified atmosphere packed foods and due to its persistence in food processing premises [6]. Although thermal treatment at temperatures > 65 °C is effective in killing *L. monocytogenes*, all cooked meats can become contaminated with listeriae during slicing and further handling. Thus, it is not surprising that not only unheated RTE foods (e.g., dry-cured or cold-smoked foods) have been identified as source for food-borne listeriosis [7] but also pasteurized products that are portioned and packed. Post-processing contamination of an otherwise nearly sterile product and prolonged shelf life under refrigerated conditions contribute to a risk scenario for introduction and multiplication of *L. monocytogenes* [8]. RTE foods implicated in food-borne listeriosis outbreaks are often of traditional type and manufactured by small local producers [9,10]. In 1993, for example, listeriosis outbreaks in France were associated with the consumption of rillettes (an RTE delicatessen food with ham cooked in grease) [11]. Besides the direct negative consequences for the health of consumers, contamination with *Listeria* requires ceasing delivery or recalls of food batches, which impairs development of domestic producers [12].

Ferreira et al. [13] reported that 50% of human listeriosis cases in the US were linked to the consumption of ready meals and that contamination was found at the retail level. In Europe, the number of listeriosis cases was found to be alarming [14]. In the European Union (EU), *L. monocytogenes* was the most serious zoonotic food-related disease with the highest fatality rate [15]. Out of 1876 cases of listeriosis, 780 were hospitalized and 167 died [15]. In November 2022, an RTE product (fish cake) from Denmark caused listeriosis in seven people (up to the publication of this manuscript, there was no further follow-up information available) [16].

In 2021, a food-borne outbreak caused by *L. monocytogenes* was reported in Austria. Five people were affected and two fatalities were noted caused by contaminated meat and meat products. Due to such cases and given the fact that every year up to two outbreaks of food-borne listeriosis are reported in Austria, 3835 samples were examined for the presence of listeriae in the year 2021, including 1300 samples of RTE food. Two of these were harmful to health and three were judged unfit for human consumption [17].

According to EU legislation [18], levels of *L. monocytogenes* in RTE foods must not exceed 100 cfu/g throughout the product’s shelf life. At the end of the manufacturing process, before the food item leaves the processing plant, the food business operator has to assure that *L. monocytogenes* is not detectable in 5 × 25 g food. For RTE foods that are considered not to support growth of *L. monocytogenes*, a limit of 100 cfu/g applies. This latter category comprises products (i) with pH ≤ 4.4 or a_w_ ≤ 0.92, (ii) with pH ≤ 5.0 and a_w_ ≤ 0.94 or (iii) with a shelf life of less than 5 days [18].

In Austria, there is a large number of RTE traditional specialties made from cured, boiled, chopped meat [19]. The standards of identity in the Austrian Food Codex [20] give no requirements in terms of pH or a_w_ for those dishes. Meat is very popular in Austria and often finds a place on the dining table at home. The Agricultural Marketing Agency (AMA) reported that in 2020, the per capita consumption of meat (including poultry) in this country was 90.8 kg, ca. 50% of which was consumed as sausages and other specialties [21].

## 2. Rationale for Application of CAP to Cooked and/or Cured Meat Products

‘Plasma’ designates a gas where a fraction of the particles is in an ionized state. This can be accomplished under various conditions, e.g., by exposing gases to an electrical field under atmospheric pressure [22]. The array of plasma species is, among others, dependent on the gases used. When ambient air is used, reactive oxygen species dominate al lower electrical voltage [23], whereas at higher voltages (10 kV), more reactive oxygen/nitrogen species (RNS, RONS) are formed [24]. The antibacterial effect of cold, atmospheric plasma (CAP) on cured meats has already been documented in a number of studies [25,26,27,28].

Generation of NO_x_ in plasma-treated water has received much attention, since nitrate will accumulate and such treated water allows curing of meats without the addition of nitrite salt [29,30,31,32,33]. Nitrite/nitrate curing of foods serves (besides other effects) as a protection against microorganisms [34,35]. However, recontamination can occur during further processing, such as shredding, portioning and packaging. Unless pH and/or water activity are sufficiently low, *L. monocytogenes* will be able to thrive in these contaminated foods [18]. A study on typical Austrian cooked ready-to-eat meat products addressed this issue in detail [36] and presented a decision tool to estimate to what extent pH or water activity of a given product need to be lowered to render a food that does not favour the multiplication of *L. monocytogenes*. The authors concluded that few, if any, options exist to lower water activity or pH without changing the sensory characteristics and impacting on acceptance of consumers.

Given the abovementioned constraints in post-processing control of *L. monocytogenes* and in consideration of the mode of action of CAP on contaminant bacteria on food surfaces, we studied the potential of CAP for reducing numbers of contaminant bacteria on two brands (‘A’, ‘B’) of sliced, cooked, cured ham (ham ‘A’, ham ‘B’), on two brands of sliced/pasteurized emulsified sausages (sausage ‘A’, sausage ‘B’) and on veal pie and calf liver pâté. We used not only *L. monocytogenes* isolates but also *L. innocua* as a surrogate [25] and *E. coli* as an established marker of (faecal) contamination [37].

## 3. Materials and Methods

### 3.1. Characterisation of the Samples and Exposure to CAP

Cured sliced meats and non-cured meat pies and pâtés were obtained pre-packed and had a shelf life > 5 days.

Samples were exposed to CAP generated by a surface barrier discharge (SBD) plasma generator (described in [38]), with two power settings (see Table 1), placed at a distance of 15 mm from the product and an exposure time of 2 and 5 min (cooked cured ham and sausages) or 3 and 5 min (non-cured meat pie and pâté). For technical reasons, 15 mm was the nearest distance we could go without running the risk of the CAP device being contaminated by contact with samples containing listeriae or *E. coli*. Selection of exposure times was based on assessment of sample colour changes during CAP exposure.

### 3.2. Measurement of Water Activity and pH

Water activity (a_w_) (Lab-Swift, Novasina, Lachen, Switzerland) and pH (penetrating electrode LoT 406-M6-DXK-S7/25; Mettler-Toledo, Urdorf, Switzerland, and pH-Meter Testo 230; Testo AG, Lenzkirch, Germany) were measured and the average of five such measurements reported.

### 3.3. Colour Measurement

Colour (L*, a*, b*) was measured in the centre of the sample’s surface using a double-beam spectrophotometer with an aperture size of 8 mm and a D65 illuminant and an observer angle of 10° (Phyma Codec 400, Phyma, Gießhübl, Austria). Surface colour was measured immediately before and after CAP treatment. Control samples (i.e., stored at ambient air under ambient light without CAP exposure) were measured at the same time intervals. Each measurement was the average of five scans and the number of replicates was 4–5. The total number of samples per product was 28 for ham and sausage, and 30 for pie and pâté.

Delta-E [(ΔE = (ΔL*)^2^ + (Δa*)^2^ + (Δb*)^2^)^0.5^] [39] was used as a proxy for visually perceived colour changes. ΔE is a single number that represents the ‘distance’ between two colours, the idea being that a ΔE of 1 is the smallest colour difference the human eye can perceive [40]. More specifically, ΔE < 2 indicates a colour change visible to an experienced observer only and ΔE > 5 indicates the impression of two different colours [41].

### 3.4. Preparation of the Inoculum and the Samples

A second set of samples was contaminated with *E. coli* (mix of NCTC 9001 and ATCC 11303), *Listeria monocytogenes* (NCTC 11994 and in-house isolate 17001) and *L. innocua* (in-house isolates 16777 and 16908-2). *E. coli* had been stored on slant agar and was activated by overnight incubation in buffered peptone water (Oxoid CM1049; Oxoid, Basingstoke, UK) at 37 °C. Likewise, freeze-dried pellets of *L. monocytogenes* and *L. innocua* were separately inoculated into brain–heart infusion broth (Merck 110493; Merck KG, Darmstadt, Germany) and incubated overnight at 30 °C. Serial decimal dilutions from the overnight cultures were prepared in 0.89% sterile saline. Aliquots from the dilutions were streaked onto plate count agar (PCA; Merck 105463), colonies were counted after 24 h incubation at 37 °C, and cell concentration/mL was calculated. In the meantime, 1:10 dilutions of the overnight cultures were maintained at 0–2 °C. This dilution was then adjusted to 7 and 6 log cfu/mL for *E. coli* and *Listeria* species, respectively. Adjusted dilutions were mixed and used within 3 h.

Samples were cut using a sterile 30 mm cork borer. On each sample surface, 100 µL or 20 µL of the mix was evenly spread. It was observed that 20 µL inoculum was easily spread on the surface (i.e., the area of the sample facing the mesh electrode of the CAP generator, taking care that there was no drip to the unexposed sides of the sample), whereas for the 100 µL inoculum, a moisture film remained. After a period of 5 min, samples were either directly vacuum-packed (control group) or exposed to CAP and then vacuum-packed (treatment group). Ham and sausage samples were stored for 24 h in the dark at 2 ± 2 °C. Veal pie and calf liver pâté samples stored for 1 and 7 days. Subsequently, the entirety of the samples was suspended in 9 parts of maximum recovery diluent (Oxoid CM0733) and macerated in a Stomacher lab blender (Seward Medical, Worthing, UK) for 3 min. Serial decimal dilutions were plated onto Listeria-selective agar (OCLA; Oxoid CM1080; incubation 72 h at 37 °C; with a turbid halo around a colony indicative of *L. monocytogenes*) and on Chrom ID E. coli agar (BioMerieux 42017; BioMerieux, Marcy l’Etoile, F; incubation 24 h at 42 °C). After incubation, typical colonies were counted and results given as log cfu/g. Experiments were done in triplicate, with a total number of samples of n = 18 per product.

### 3.5. Statistical Analysis

The colour values before and after treatment were analysed by pairwise comparison (paired *t*-test), with a level of significance set to *p* < 0.05. Within each product group (cured ham; cured sausage; non-cured meats), water activity and pH of the two samples each were compared by t-test. For each storage day, numbers of bacteria in CAP-exposed samples were compared to those in the control samples (multiple sample comparison procedure; Statgraphics 3.0, Statistical Graphics Corp., Warrenton, VA, USA), with a level of significance set to *p* < 0.05.

## 4. Results

### 4.1. Water Activity and pH of Samples

Water activity and pH of samples are reported in Table 2. According to current EU legislation [18], samples were considered to be able to support growth of *L. monocytogenes*.

Statistically significant, yet small differences were observed for pH between the two sausage samples, and for pH and water activity between the two non-cured meats.

### 4.2. Changes in Colour

After CAP treatment of cooked cured ham ‘A’, a statistically significant decrease in a* values was observed immediately following 2 min CAP exposure at low power, and after 5 min exposure to high and low power. An increase of b* was observed after 5 min exposure. ΔE values were in the range of 1.4 to 2.1. No statistically significant changes in colour parameters were observed in controls (Table 3). In ham ‘B’ (high power), statistically significant changes in a* were observed only after exposure to high power CAP. As for ham ‘A’, an increase of b* was observed after 5 min exposure, and no statistically significant changes in colour parameters were observed in controls. ΔE values were in the range of 1.2 to 1.9 (Table 3).

For the two sliced cured sausage samples, a statistically significant decrease in a* values was observed immediately following CAP exposure, regardless of the mode of CAP exposure protocol and of sample type (Table 4), whereas no significant decrease was observed in non–CAP-exposed controls. Likewise, a significant, yet small increase in b* values was observed in CAP-exposed sausage ‘A’ samples and in sausage ‘B’ at 5 min. However, the differences were small, and did not exceed 1 for a* and b* at 2 min exposure or 3 at 5 min. exposure. Average ΔE values were in the range of 0.9 to 1.3 for 2 min exposure, and slightly higher after 5 min exposure (1.6–2.7).

Similar findings were found for CAP-exposed veal pie and calf liver pâté, with average ΔE values in the range of 0.7–1.7 and 1.1–2.0, respectively (Table 5). In all treatments, a small, yet significant increase was found for redness (a*). In liver pâté, lightness (L*) decreased significantly. In control samples exposed to air and ambient light, no significant differences were observed.

Although ΔE values > 2 indicate changes in colour visible also for inexperienced observers, it is assumed that a change in colour is perceived by the majority of consumers at higher ΔE values of >3 [42]. Thus, we used the 5 min exposure protocol for subsequent experiments with bacterial contaminants.

### 4.3. Changes in Bacterial Load

Numbers of *E. coli* were significantly lower (*p* < 0.05) in CAP-exposed cured sausage and ham than in controls (Figure 1). Significant reductions of numbers of listeriae in CAP-exposed samples were found in cured ham ‘B’, and in sliced sausage samples ‘A’ and ‘B’ (only 20 µL inoculum). For the sake of simplicity, only *L. monocytogenes* will be reported in the following, since we observed the same ratio between *L. monocytogenes* and *L. innocua* in the inoculum as well as on controls and CAP-exposed samples.

CAP was obviously more effective in inactivating *E. coli* (1 to 3 log cycles) than *Listeria* (from 0.2 to max. 1.5 log cycles; sliced sausage ‘A’). There was no consistent pattern as regards the effect of inoculum size and plasma type (low power or high power).

In veal pie and calf liver pâté, no significant reductions were observed for *E. coli* 24 h after CAP exposure, whereas after 7 days’ storage, a significant reduction was observed only for samples exposed to low-power CAP (up to 1 log cycle). In veal pie, levels of *Listeria* were significantly reduced in samples tested at 24 h (at level of ca. 0.6 log cycles), but not in calf liver pâté. Significant reductions in listeriae were observed only in liver pâté 7 days after low-power CAP exposure (ca. 0.4 log cycles; Figure 2).

## 5. Discussion

### 5.1. Effect on Contaminant Bacteria

We considered typical cured and non-cured, heat-treated, ready-to-eat meat pro-ducts that can easily be contaminated during portioning and slicing. Physicochemical characteristics indicated that the products can favour the multiplication of *Listeria monocytogenes* during the shelf life of these products [18]. For such high-risk products, strict adherence to good hygiene practices is a prerequisite, and the establishment of operation prerequisite programs should be considered [43].

The implementation of additional antibacterial measures/interventions has been suggested repeatedly, but the magnitude of the effect of biological agents is not always certain (e.g., anti-listerial bacteriophages [44,45]) and limitations may apply to physicochemical treatments in terms of residues or changes of organoleptic properties of properties (see EFSA series of scientific opinions).

Surface pasteurization of vacuum-packed cooked ready-to-eat meat products requires temperatures of 96 °C and holding times of 10 min to effectuate a 2 to 4 log reduction in *Listeria monocytogenes* [46], but such conditions are not feasible for all meat products. A 1% cetylpyridinium chloride (CPC) spray applied on Polish-style sausage before vacuum-packing was highly effective against *L. monocytogenes* (depending on the inoculation level, an immediate reduction of 1–3 log cycles was observed, and after 42 days of storage it was 2–4 log units) [47], but such additives are not accepted by all consumers.

Cold atmospheric plasma has demonstrated its ability to reduce numbers of bacteria on food surfaces and is thus well suited for managing bacterial contamination post-processing and pre-packaging [25]. With regard to ready-to-eat meats, the reductions of *Listeria* we observed (up to 1.5 log) are in the range as reported in other studies [25,26,27,28], albeit differences in experimental design make detailed comparisons difficult.

Our results support the assumption that Gram-negative bacteria (*E. coli*) are more susceptible to CAP than are Gram-positives (*Listeria*). This assumption would be logical since the membrane lipids in Gram-negative organisms are directly exposed to CAP molecules in particular ozone, whereas the cell wall of Gram-positive organisms would protect the cell membrane [48]. However, experimental data are inconclusive [26,37]. While the higher susceptibility of *E. coli* could be explained, it is unclear why *Listeria* reduction differed between similar products (‘A’, ‘B’) from different producers, the more so as the ingredient list was nearly identical. The lower reductions observed in pâté and pie compared to cured meats deserves attention and warrants further studies, particularly as on the labels of these products, no antioxidants were declared. We observed no consistent pattern as regards the effect of inoculum size (with respect of the moisture film on the sample surface) or plasma type (low power or high power), although it has been established that humidity or water films influence plasma composition [26,49,50] and that ROS and RONS act differently on bacterial cells [48]. Since our studies were designed as pilots, further experiments are envisaged to study these issues in detail.

A limitation of the methodology we applied for enumeration of bacteria after CAP exposure is that direct plating onto selective agar media for enumeration of bacteria does not consider the possibility of sublethal injury or a viable but not culturable (VBNC) state of the contaminant bacteria [51] post-CAP exposure. The VBNC issue has been studied for thermal and acidic stress in *Listeria* [52,53,54,55], but specific studies on CAP are still lacking.

Likewise, instead of the pre-packaging CAP exposure we studied, an in-package CAP treatment with formation of the plasma species in the headspace of the package could be more feasible, since the product is then already sealed and protected from contamination [25,56].

### 5.2. Effect on the Food Matrix

The role of the food matrix in the CAP–bacterium interplay is poorly studied. It can be expected that-given the abundance of meat protein, fat and water in the food matrix compared to that in the bacterial cells, the majority of CAP species react with the food matrix. As regards plasma generated from ambient air, it is debatable if reactive oxygen substances and reactive nitrogen substances in the plasma would react with the food matrix in a way that would result in a ‘novel food’, i.e., in molecular structures that were not present in foods within the EU before 15 May 1997 (Article 3 of Regulation (EU) 2015/2283 [57]). With respect to muscle foods, we recently reviewed the effect of CAP on myoglobin forms and thus on colour [58]. In the cured meat products, veal pie and calf liver pâté, a decrease in redness (a*) was most frequently observed, indicating some effect on the myoglobin forms present in the meat products, and in fewer cases a significant, yet small increase in b*. Lightness (L*) was not affected at all, indicating that CAP exposure had no effect on water-binding capacity.

The magnitude of changes in a* and b* was moderate: ΔE values of up to 2.2 were observed for some 2 min exposure protocols, and values up to 2.7 for 5 min exposure. Notably, in the control samples, ΔE values were consistently <1, whereas in all treatment groups, it was >1. ΔE values < 1 are not likely to be recognised as differences, and values of >2 indicate changes in colour also visible to inexperienced observers, and it is assumed that a change in colour is perceived by the majority of consumers at ΔE values of >3 [39,40,41,42]. Further experiments should explore if or to what extent colour differences are observed in CAP-exposed samples after cold storage.

The small increase in redness (a*) in non-cured products is most probably not due to a curing reaction, since the myoglobin is already heat-denaturated. A decrease in lightness in CAP-exposed liver pâté might simply indicate that the product is more sensitive to drying [59] than other products under study.

## 6. Conclusions

CAP treatment of sliced, cured, cooked ham and sausage effectuated significant reductions in *E. coli* (up to 3 log units), but less pronounced reductions in listeriae. In traditional non-cured cooked meats, CAP was less effective. Colour changes (expressed as ΔE values) were in an acceptable range, although changes in redness (a*) indicated some effect of CAP on the myoglobin present in cured foods. Differences were observed between cured and non-cured meats, but also between products of similar type, which warrants further studies.

## Figures and Tables

**Figure 1 foods-12-00685-f001:**
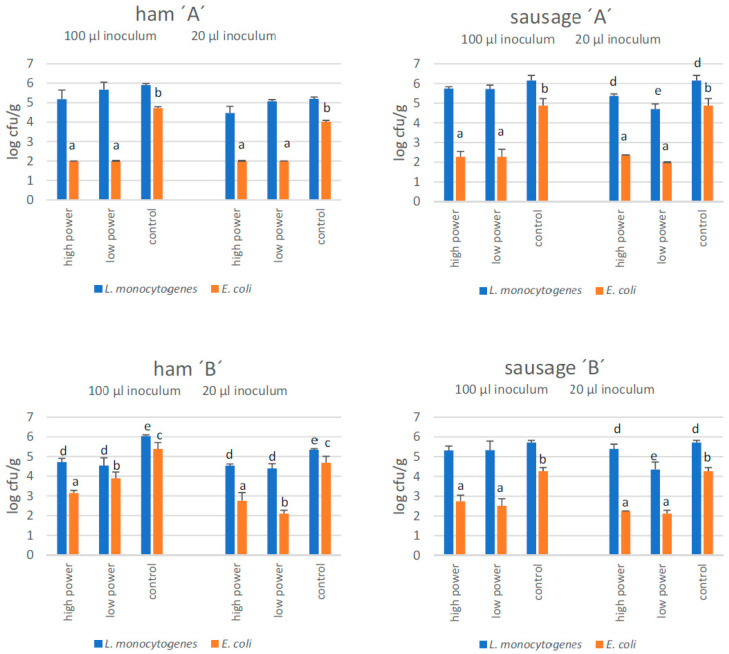
The effects of a 5 min exposure to high-power vs. low-power cold atmospheric pressure plasma treatment on the survival of *L(isteria)* and *E(scherichia) coli*, inoculated on the surface of cured/cooked/sliced meat products (n = 3) manufactured in two different enterprises (‘A’ and ‘B’). Note that the limit of detection is 2.0 log cfu/g, i.e., bars at 2.0 log with a standard deviation of 0 indicate that bacterial counts were actually <2 log. Within the 100 µL or 20 µL inoculum groups, statistically significant differences (*p* < 0.05) between numbers of bacteria on treated and control samples are indicated by different superscripts (a,b,c for *E. coli* and d,e, for *L. monocytogenes*) above the columns.

**Figure 2 foods-12-00685-f002:**
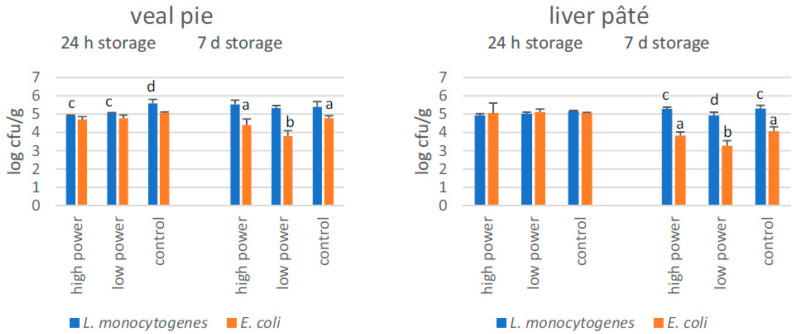
Numbers of *L(isteria) monocytogenes* and *E(scherichia) coli* inoculated on veal pie and calf liver pâté (n = 3) and subjected to 5 min exposure to high- vs. low-power cold atmospheric pressure plasma, followed by 1 to 7 days’ storage at 2 ± 2 °C. For the 1- and 7-day storage groups, statistically significant differences (*p* < 0.05) between numbers of bacteria on treated and control samples are indicated by different superscripts (a,b for *E. coli* and c,d for *L. monocytogenes*) above the columns.

**Table 1 foods-12-00685-t001:** Settings of the CAP device.

		Comment
CAP device	SBD-type, 9 kHz frequency	Device described in Bauer et al. [38]
CAP settings	low power	Power input 20.7 W Output voltage 8.16 kV Power density 0.48 W/cm^2^
	high power	Power input 29.9 W Output voltage 9.44 kV Power density 0.67 W/cm^2^
Exposure time	2 or 5 min	for cooked cured ham and cooked cured sausage
	3 or 5 min	for veal pie and calf liver pâté
Distance sample to electrode	15 mm	for all samples

**Table 2 foods-12-00685-t002:** Physicochemical sample characteristics.

		Characteristics	
Product	Code	pH (n = 5)	Water Activity (a_w_) (n = 5)
Sliced cooked cured ham	Ham ‘A *’ Ham ‘B *’	6.28 ± 0.03 6.32 ± 0.02	0.96 ± 0.01 0.96 ± 0.01
Sliced cooked cured sausage	Sausage ‘A *’ Sausage ‘B *’	6.27 ^a^ ** ± 0.02 6.33 ^b^ ± 0.02	0.95 ± 0.01 0.96 ± 0.01
Sliced cooked meats	Veal pie Calf liver pâté	5.95 ^a^ ± 0.01 5.48 ^b^ ± 0.04	0.92 ^c^ ± 0.01 0.94 ^d^ ± 0.01

* ‘A’ and ‘B’ indicate the manufacturer. ** Figures with different superscripts differ significantly (*p* < 0.05).

**Table 3 foods-12-00685-t003:** Colour (L*, a* and b*) of cooked cured ham before and after CAP exposure, with colour difference expressed as ΔE.

Brand	Power	Time (min)	Prior to or after Treatment	L*	a*	b*	ΔE
A	low	2	P	72.8 ± 0.9	4.0 ^a^ ± 0.4	8.3 ± 0.8	
			a	74.2 ± 1.9	2.4 ^b^ ± 0.8	8.4 ± 0.5	2.11
	high	2	P	71.9 ± 2.8	4.7 ± 1.0	8.9 ± 0.5	
			a	70.2 ± 1.0	4.4 ± 0.7	8.8 ± 0.4	1.75
	no (control)	2	P	72.0 ± 1.1	4.2 ± 0.5	8.5 ± 0.9	
			a	71.6 ± 1.4	4.1 ± 0.6	8.6 ± 0.8	0.42
	low	5	P	71.0 ± 3.3	5.3 ^a^ ± 1.3	7.3 ^c^ ± 0.3	
			a	70.8 ± 2.4	4.2 ^b^ ± 1.1	8.5 ^d^ ± 0.2	1.61
	high	5	P	71.3 ± 2.4	4.7 ^a^ ± 0.8	8.7 ^c^ ± 1.1	
			a	71.5 ± 2.6	3.9 ^b^ ± 0.8	9.9 ^d^ ± 1.3	1.39
	no (control)	5	P	72.0 ± 1.1	4.2 ± 0.5	8.5 ± 0.9	
			a	71.3 ± 1.5	4.0 ± 0.8	8.36 ± 0.7	0.74
B	low	2	P	68.2 ± 2.3	8.0 ± 1.0	8.6 ± 0.3	
			a	67.5 ± 2.8	7.0 ± 1.3	8.6 ± 1.0	1.21
	high	2	P	68.6 ± 2.1	6.8 ^a^ ± 1.0	8.5 ± 1.0	
			a	69.4 ± 2.0	5.9 ^b^ ± 0.9	8.9 ± 1.0	1.18
	no (control)	2	P	71.0 ± 1.6	7.3 ± 1.3	7.1 ± 0.5	
			a	71.1 ± 1,9	6.9 ± 0.9	7.3 ± 0.3	0.46
	low	5	P	70.3 ± 3.5	7.0 ± 1.5	7.0 ^c^ ± 0.3	
			a	70.2 ± 3.4	6.2 ± 1.3	8.1 ^d^ ± 0.6	1.35
	high	5	P	71.0 ± 2.8	7.3 ^a^ ± 1.2	7.5 ^c^ ± 0.6	
			a	71.5 ± 2.0	6.0 ^b^ ± 0.9	8.8 ^d^ ± 0.3	1.85
	no (control)	5	P	71.0 ± 1.6	7.3 ± 1.3	7.1 ± 0.5	
			a	71.2 ± 1,5	6.7 ± 1.5	7.2 ± 0.6	0.64

Note: n = 5 for low- and n = 4 for high-power treatment. Within-sample treatment combinations, different superscripts indicate statistically significant (*p* < 0.05) differences between colour parameters before and after treatment.

**Table 4 foods-12-00685-t004:** Colour (L*, a* and b*) of cooked cured sausage before and after CAP exposure, with colour difference expressed as ΔE.

Brand	Power	Time (min)	Prior to or after Treatment	L*	a*	b*	ΔE
A	low	2	P	74.8 ± 0.6	6.4 ^a^ ± 0.3	9.7 ^c^ ± 0.5	
			a	74.9 ± 1.5	5.6 ^b^ ± 0.2	10.2 ^d^ ± 0.5	0.89
	high	2	P	74.8 ± 1.0	6.5 ^a^ ± 0.3	9.8 ^c^ ± 0.6	
			a	73.5 ± 1.9	6.0 ^b^ ± 0.2	10.7 ^d^ ± 0.6	1.64
	no (control)	2	P	74.5 ± 0.6	6.7 ± 0.5	9.0 ± 0.5	
			a	74.1 ± 0.9	6.8 ± 0.6	9.3 ± 0.4	0.51
	low	5	P	74.1 ± 0.7	6.5 ^a^ ± 0.4	9.7 ^c^ ± 0.1	
			a	74.3 ± 0.6	5.5 ^b^ ± 0.3	10.3 ^d^ ± 0.1	1.19
	high	5	P	74.1 ± 1.6	7.0 ^a^ ± 0.1	8.9 ^c^ ± 0.2	
			a	73.7 ± 0.8	5.3 ^b^ ± 0.1	10.1 ^d^ ± 0.4	2.16
	no (control)	2	P	74.5 ± 0.6	6.7 ± 0.5	9.0 ± 0.5	
			a	73.9 ± 1.0	7.0 ± 0.8	9.2 ± 0.6	0.71
B	low	2	P	66.2 ± 1.0	11.9 ^a^ ± 0.7	9.6 ± 0.8	
			a	65.4 ± 1.1	10.8 ^b^ ± 0.8	9.7 ± 0.6	1.33
	high	2	P	66.1 ± 0.8	12.4 ^a^ ± 0.5	9.6 ± 0.6	
			a	65.5 ± 1.3	11.8 ^b^ ± 0.7	9.7 ± 0.8	0.89
	no (control)	5	P	64.0 ± 1.2	13.2 ^a^ ± 0.7	8.7 ± 0.4	
			a	64.6 ± 1.2	12.9 ^b^ ± 0.4	9.0 ± 0.5	0.73
	low	5	P	64.3 ± 1.0	13.3 ^a^ ± 0.6	8.5 ^c^ ± 0.2	
			a	64.8 ± 1.3	10.6 ^b^ ± 0.2	9.1 ^d^ ± 0.3	2.73
	high	5	P	65.4 ± 1.1	13.3 ^a^ ± 0.2	8.6 ^c^ ± 0.4	
			a	65.2 ± 0.5	10.7 ^b^ ± 0.4	9.4 ^d^ ± 0.3	2.72
	no (control)	5	P	64.0 ± 1.2	13.2 ± 0.7	8.7 ± 0.4	
			a	63.1 ± 1.7	12.7 ± 1.0	8.9 ± 0.7	0.81

Note: n = 5 for low- and n = 4 for high-power treatment. Within-sample treatment combinations, different superscripts indicate statistically significant (*p* < 0.05) differences between colour parameters before and after treatment.

**Table 5 foods-12-00685-t005:** Colour (L*, a* and b*) of (non-cured) veal pie and liver pâté before and after CAP exposure, with colour difference expressed as ΔE.

Product	Power	Time (min)	Prior to or after Treatment	L*	a*	b*	ΔE
Veal pie	low	3	P	57.5 ± 0.5	12.9 ^c^ ± 0.2	15.0 ± 0.2	
			a	56.8 ± 0.7	13.8 ^d^ ± 0.2	15.3 ± 0.2	1.21
	high	3	P	57.9 ± 0.5	13.1 ^c^ ± 0.2	15.2 ± 0.2	
			a	57.5 ± 0.4	14.8 ^d^ ± 0.2	15.5 ± 0.3	1.68
	no (control)	3	P	57.9 ± 0.5	13.1 ± 0.2	15.2 ± 0.3	
			a	57.8 ± 0.5	13.6 ± 0.7	15.5 ± 0.3	0.58
	low	5	P	56.6 ± 0.9	13.5 ^c^ ± 0.4	15.2 ± 0.2	
			a	56.1 ± 0.7	14.5 ^d^ ± 0.1	14.9 ± 0.1	1.12
	high	5	P	57.0 ± 0.8	13.1 ^c^ ± 0.2	14.7 ± 0.4	
			a	57.2 ± 0.8	13.7 ^d^ ± 0.4	15.0 ± 0.2	0.74
	no (control)	5	P	57.9 ± 0.5	13.1 ± 0.2	15.2 ± 0.3	
			a	57.7 ± 0.4	13.5 ± 0.7	15.5 ± 0.3	0.52
Liver pâté	low	3	P	63.9 ^a^ ± 1.6	12.2 ^c^ ± 0.5	15.5 ± 0.5	
			a	62.9 ^b^ ± 1.2	12.6 ^d^ ± 0.6	15.9 ± 0.7	1.10
	high	3	P	63.3 ^a^ ± 0.6	11.9 ^c^ ± 0.2	15.8 ± 0.3	
			a	62.5 ^b^ ± 0.5	12.7 ^d^ ± 0.5	16.4 ± 0.3	1.28
	no (control)	3	P	63.9 ± 0.6	12.5 ± 0.3	15.6 ± 0.4	
			a	63.5 ± 0.7	12.6 ± 0.3	16.0 ± 0.6	0.58
	low	5	P	64.0 ^a^ ± 0.7	12.1 ^c^ ± 0.3	15.6 ± 0.4	
			a	63.1 ^b^ ± 0.4	13.1 ^d^ ± 0.3	16.1 ± 0.6	1,39
	high	5	P	63.0 ^a^ ± 1.0	12.5 ^c^ ± 0.3	16.0 ± 0.4	
			a	61.4 ^b^ ± 1.1	13.6 ^d^ ± 0.3	16.4 ± 0.5	1.96
	no (control)	5	P	63.9 ± 0.6	12.5 ± 0.3	15.6 ± 0.4	
			a	63.5 ± 0.5	12.6 ± 0.4	16.1 ± 0.6	0.61

Note: n = 5 for low- and high-power treatment. Within-sample treatment combinations, different superscripts indicate statistically significant (*p* < 0.05) differences between colour parameters before and after treatment.

## Data Availability

Data are contained in the DVM Diploma Theses from Ute Vankat, Julia Schrei and Michelle Graf. These are available at https://permalink.obvsg.at/AC15668870 (accessed on 24 January 2023), https://permalink.obvsg.at/AC15615153 (accessed on 24 January 2023), and https://permalink.obvsg.at/AC15615235 (accessed on 24 January 2023).

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
