# Peer review of "Treatment of Ready-To-Eat Cooked Meat Products with Cold Atmospheric Plasma to Inactivate Listeria and Escherichia coli"

_foods, 2023, doi:10.3390/foods12040685_

Round 1

Reviewer 1 Report

The manuscript described an interesting study about the CAP systems in RTE meat products. It is evident that the use of CAP in ready-to-eat meat products will contribute to the literature in terms of its effect on food safety. I think that further detailing the research design of the study will contribute to future studies.

Therefore, I make the following recommendations.

It would be better if your trial design indicated how many samples were used for each meat product.

Line 131: “placed at a distance of 15 mm from the product and an exposure time of 2 and 5 minutes (cooked cured ham and sausages) or 3 and 5 minutes (non-cured meat pie and pâté).” is there any specific reason to choose this parameters? 

Author Response

Thank you for your comments. Please find our changes and responses in the attached document. Sincerely, Isabella CSadek (on behalf of F Smulders)

Reviewer 2 Report

The manuscript evaluates the treatment of ready-to-eat cooked meat products with cold atmospheric plasma to inactivate Listeria and E. coli.

The protocols were well detailed; the results were well showed and the references were current.

I suggest more discussion of the results with other works.

Author Response

(The authors gave the same response as above.)

Reviewer 3 Report

The paper deals with the application of cold plasma to food as a measure to extend shelf life by non-thermal preservation methods with emphasis on microbiological and colour aspects.

The paper is well organised and has no major shortcomings. Some opportunities for improvement/comments are mentioned below.

- In the abstract it is assumed that ready-to-eat products would be a vehicle for listeria, consider a potential vehicle, as it is not possible to be so categorical.

- The summary does not specify the conditions for the application of KAP, only the timing. The document would be improved if it also presented the missing information.

In l27 avoid mentioning that it would obviously be more effective to apply CAP.

- Check if comma or dot is used as a separator of thousands l72 and l79.

Revise the use of the sub-index to refer to water activity.

Revise the wording of l115-l116.

l131 To improve the understanding and reproducibility of the experiments, it is recommended to adequately describe the devices used for the application of cold plasma and the set-ups.

Table 1 review the relevance and/or improve the presentation of the table, as it is confusing.

l147 missing the information about the observer in °, also define where these five measurements were performed (according to some pattern, in the centre, at the ends for example).

In table 2 there is no heading in the first column.

Check the scale of figure 2, in order not to fall or induce visual errors, it would be better if figures 1 and 2 had the same scale.

The illustration could be improved, to give an adequate account of the contribution of the work to knowledge in the discipline of food preservation by non-thermal methods such as cold plasma.

Revise the format of reference 11 (remove the underlined DOI).

Author Response

(The authors gave the same response as above.)
